# Fundamental Shifts in the EU's Electric Power Sector Development: LMDI Decomposition Analysis

Viktor Koval [1,*], Viktoriia Khaustova [2], Stella Lippolis [3], Olha Ilyash [4], Tetiana Salashenko [2] and Piotr Olczak [5]

[1] Department of Business and Tourism Management, Izmail State University of Humanities, 12 Repina Str., 68601 Izmail, Ukraine

[2] Research Centre for Industrial Problems of Development, National Academy of Sciences of Ukraine, 1a Inzhenernyi Ln., 61166 Kharkiv, Ukraine; v.khaust@gmail.com (V.K.); tisandch@gmail.com (T.S.)

[3] Ionian Department of Law, Economics and Environment, University of Bari Aldo Moro, Via Duomo 259, 74100 Taranto, Italy; stella.lippolis@uniba.it

[4] Igor Sikorsky Kyiv Polytechnic Institute, National Technical University of Ukraine, 37 Prosp. Peremohy, 03056 Kyiv, Ukraine; oliai@meta.ua

[5] Mineral and Energy Economy Research Institute of the Polish Academy of Sciences, 7A Wybickiego Street, 31-261 Kraków, Poland; olczak@min-pan.krakow.pl

* Correspondence: victor-koval@ukr.net

**Abstract:** The electric power sector plays a central role in changing the EU's energy landscape and establishing Europe as the first climate-neutral continent in the world. This paper investigates fundamental shifts in the EU's electric power sector by carrying out its logarithmic mean Divisia index decomposition by stages of electricity flows on a large-scale basis (for both the entire EU and its 25 member states) for the period 1995–2021 and identifies the individual contribution of each EU member state to these shifts. In this study, four decomposition models were proposed and 14 impact factors (extensive, structural, and intensive) affecting the development of the EU electric power sector were evaluated in absolute and relative terms. It was found that the wind–gas transition, which took place in the EU's electric power sector, was accompanied by an increase in the transformation efficiency of inputs in electricity generation and a drop in the intensity of final energy consumption. The non-industrial reorientation of the EU's economy also resulted in a decrease in the final electricity consumption. At the same time, this transition led to negative shifts in the structure and utilization of its generation capacities. The fundamental shifts occurred mainly at the expense of large economies (Germany, France, Spain, and Italy), but smaller economies (Romania, Poland, Croatia, the Netherlands, and others) made significant efforts to accelerate them, although their contributions on a pan-European scale were less tangible.

**Keywords:** electric power sector; LMDI decomposition; impact factors; efficiency; structural shifts; electricity flows; energy transition; EU

## 1. Introduction

The EU is striving to change its energy landscape and make Europe the first climate-neutral continent in the world by 2050 [1]. To this end, tectonic fundamental shifts, where the central place belongs to the transformation of the electric power sector, are being implemented. The idea of rebuilding the EU's electric power sector is not new; it was first enshrined in the EU Commission Working Document "The Internal Energy Market" (1988) and gradually progressed, with targets steadily raised through the adoption of the EU's First (1996), Second (2003), Third (2009), and Fourth (2018–2019) Energy Packages [2–6].

The last one, the so-called Winter Package, "Clean Energy for All Europeans", includes three legislative documents that focus directly on electricity issues, while others consider electricity as an important part of the energy transition of Europe as a whole [6]. Since the

start of the Russian invasion of Ukraine, the European Commission has raised its targets again. Thus, at present, according to the RePowerEU Plan, it is expected to ensure the generation of renewable energy at a level of 45% by 2030 [7]. This goal does not seem too ambitious, given that the share of RESs in total electricity generation increased from 14% in 1995 to 35% in 2021 [8]. At the same time, this challenge requires understanding the factors that have contributed to positive fundamental shifts in the electric power sector in the past in order to transpose them into the future as well as to overcome the negative impacts of others. This theoretical and applied problem constitutes the aim of this paper as a study of the fundamental shifts in the development of the EU's electric power sector over the long-term horizon. It implies an assessment of the individual contributions of EU member states to the overall shifts in the EU's electric power sector development. To solve this problem, a large-scale decomposition analysis based on the input–output model of electricity flows was chosen.

The rest of the paper is organized as follows. Section 2 examines the specifics of decomposition analysis, focusing on the shortcomings of previous research on its application to the electric power sector. Section 3 presents the proposed methodology for decomposition analysis of the electric power sector by stages of electricity flows. Section 4 focuses on the results of the decomposition analysis of the EU's electric power sector development. The discussion is presented in Section 5, and the conclusions are drawn in Section 6.

## 2. Literature Review

Developing issues in the electric power sector have received a lot of attention in research. A wide range of these problems have formed a relevant research field in economics, which includes studying the sustainability of electric power sector development [9–13], the liberalization of electricity markets [14–18], electricity pricing policy and price fluctuations [19–22], and the forecasting of energy mix and fuel consumption [23,24]. Today, the electric power sector's development is considered relevant to ensuring EU energy security [25–28] and its energy transition to decarbonization [29–35].

Decomposition analysis should be seen as one of the powerful tools for studying long-term extensive, structural, and intensive shifts in the energy sector. This approach emerged in the early 1980s [36], and today two of its forms are used in practice: structural decomposition analysis (SDA) [37] and index decomposition analysis (IDA) [38,39]. According to Hoekstra and van der Bergh (2003), the differences between them are that the former is based on input–output coefficients and final demand per sector, while the latter is based on output per sector. IDA is most commonly used to examine aggregated industry data and is also more suitable for detailed studies over time and by country [40]. Initially, IDA was performed based on the Laspeyres decomposition method [41–44], but later, researchers switched to the Divisia decomposition technique [45–49], using the arithmetic mean Divisia index (AMDI) [50,51], and to the logarithmic mean Divisia index (LMDI) method [52–54], as well as to using both of them [55]. The LMDI decomposition approach, proposed by Ang and Zhang [38], yields an almost perfect decomposition, leaving no residuals in the decomposition model compared to the AMDI decomposition approach [56]. As discovered by one of the inventors, after more than 15 years of using LMDI decomposition, it was proven to produce results with no residuals, involving many factors and spatial decomposition and integrating the physical and economic components ([36], p. 234). There are two types of LMDI models: LMDI-I and LMDI-II, which can be applied additively or multiplicatively. The results they provide are very similar, so researchers prefer to use LMDI-I models because of the ease of computational procedures [36,38,39,46].

One can find many applications of the LMDI decomposition method for large-scale energy studies ([56–60], etc.), but so far insufficient attention has been paid to decomposition analysis of the electric power sector specifically, and, in most cases, such analysis is restricted to one stage of electricity flows or one country. The Web of Science database contains 1163 publications focusing on the LMDI decomposition method, and only 108 of them deal with problems of the electric power sector's development (as of 10 May 2023) [61]. To

formalize the research field of LMDI decomposition of electric power sector development, a bibliographic analysis was conducted using VOSviewer (Appendix A), designed by the Centre for Science and Technology Studies (CWTS) of Leiden University [62]. This analysis revealed that the LMDI decomposition of electric power sector development prioritized the issue of $CO_2$ emissions, seeking to achieve carbon neutrality and mitigate climate change. The main driving factors considered included urbanization, economic growth, intensity, and efficiency, which could affect electricity consumption and production. Table 1 shows some typical studies on LMDI decomposition analysis of electric power sector development which are closely related to our study.

**Table 1.** Summary of recent studies on LMDI decomposition of electric power sector development. Source: Own study based on the WoS database [61].

| Study | Result Factor | Impact Factors | Countries | Studied Years |
|---|---|---|---|---|
| Zhongfu et al. (2011) [63] | $CO_2$ emissions in the electric power sector | $CO_2$ emission coefficient, energy intensity of power generation, power generation and consumption ratio, electricity intensity of the gross domestic product (GDP), provincial structural change, and the energy intensity of the GDP | China | 1998–2008 |
| Karmellos et al. (2016) [64] | $CO_2$ emissions in the electric power sector | Level of activity, electricity intensity, electricity trade, efficiency of electricity generation, and fuel mix | EU-28 | 2000–2012 |
| Jiang (2017) [65] | $CO_2$ emissions in the electric power sector | Electricity output effect, energy mix effect, and conversion efficiency effect | US | 1990–2014 |
| Xie et al. (2019) [66] | $CO_2$ emissions in the electric power sector | Energy consumption in power generation, thermal power structure, power generation structure, transmission and distribution loss, electrification, energy intensity, and economic scale | China | 1985–2016 |
| De Oliveira-De Jesus (2019) [67] | $CO_2$ emissions intensity of electricity | Fuel mix, thermal efficiency, fossil share, and geographical effects | Latin America and the Caribbean | 1990–2014 |
| Rüstemoğlu (2019) [68] | $CO_2$ emissions in the electric power sector | Economic activity, the fuel structure effect, the pollution effect, and electricity intensity | Germany | 1990–2015 |
| Chen et al. (2019) [69] | Solar PV electricity output | Solar system efficiency, curtailment issues, and solar resources | China and the US | 2008–2015 |
| Yu et al. (2020) [70] | Thermal power generation | Economic growth, electricity security, substitution effect, electricity intensity, and technological structure | 25 EU countries | 1997–2017 |
| Miškinis et al. (2021) [71] | $CO_2$ emissions | Impact of population change, economic growth, decline in energy intensity, RES deployment, and reduction in emissions intensity on change in GHG emissions | Baltic States | 2010–2019 |
| Sadorsky (2021) [72] | Wind energy consumption | Renewable energy share component and improvements in energy intensity | 17 European countries | 2002–2017 |
| Rivera-Niquepa et al. (2022) [73] | $CO_2$ emissions intensities of electricity generation | Carbon intensity, generation efficiency, and contribution of fossil generation at the specific and total level of electricity production | 8 Colombian administrations | 1990–2020 |
| Shi et al. (2022) [74] | $CO_2$ emissions from electricity systems | Total energy consumed in power generation stage, generation structure, carbon emissions factors by power generation and by grids, electricity consumption, GDP, and population | Gansu Province, China | 2000–2019 |
| Yu et al. (2023) [75] | Renewable electricity generation | Electricity consumption scale, electricity productivity, output productivity, technical efficiency, and carbon emissions | China | 1995–2018 |

**Table 1.** *Cont.*

| Study | Result Factor | Impact Factors | Countries | Studied Years |
|---|---|---|---|---|
| Koilakou et al. (2023) [76] | Carbon and energy intensity | Income, population, energy intensity, and energy structure | The US and Germany | 2000–2017 |
| Zhang et al. (2023) [77] | $CO_2$ emissions intensities of electricity generation | Energy structure, energy intensity, clean production, supply structure, and power loss effects | 6 Chinese regions | 2005–2020 |

Thus, there is now a lack of profound research into the simultaneous decomposition of the entire electric power sector for all stages of electricity flows; only [74] conducted such research for the Gansu province of China. Most previous studies focused on the decomposition analysis of the driving factors influencing $CO_2$ emissions from the electric power sector [63–68,71,73,74,76,77], and such analysis was rarely conducted by type of electric power generation [69,70,72,75]. In recent years, there has been an increase in the number of profound studies on decomposition analysis of electric power sector development [64,67,69–73,76,77], but we struggled to find a large-scale decomposition analysis of the EU's electric power sector; only the authors of [64] investigated it, using five impact factors to decompose $CO_2$ emissions form the electric power sector, and the authors of [70,71] developed a decomposition model of the EU's wind and thermal energy sectors, using two and five impact factors, respectively.

At the same time, shifts at each stage of electricity flows necessarily entail changes in the entire balance of the electric power sector, so that it must be maintained constantly. Conducting a decomposition of the electric power sector for all stages of electricity flows simultaneously makes it possible to comprehensively track shifts in individual impact factors. Thus, there is an objective need to deepen decomposition analysis with respect to the stages of electricity flows and conduct it on a large scale for the entire EU.

## 3. Materials and Methods

The decomposition analysis of the EU's electric power sector involved the formation of a database using the information from the Eurostat Database for 1995–2021, both for the EU as a whole and for continental EU member states. The choice of the period was conditioned by the availability of data for all components. Countries such as Luxembourg, Malta, and Cyprus were excluded due to the fact that they did not have a complete data set for the entire study period, such that their inclusion could have narrowed the long-term horizon of the study. Thus, the current research is based on the following Eurostat datasets: transformation inputs and gross electricity generation from energy flow Sankey diagram data [78]; supply, transformation, and consumption of electricity [79]; gross and net electricity generation by type of fuel [80]; electricity generation capacities by main fuel groups [81]; economic outputs by sectors [82]; and households' expenditures [83].

One of the problematic issues in conducting the decomposition analysis of the EU's electric power sector is the different classifications of various data, both by fuel type within the energy database and by type of economic activity. In some cases, these issues can be resolved by combining the data into common groups:

- Table A1 presents a grouped classification of transformation inputs and types of electricity generation according to the UN classification [84];
- Table A2 presents the combined types of economic activity according to the European Community classification [85].

In other cases, it was necessary to divide the decomposition equation into components: (i) for gross and net electricity generation, and (ii) for available electricity and final electricity consumption.

The division of the electric power sector into the stages of transformation, supply, and consumption of electricity allowed for the following equations, which formed the basis of the decomposition analysis:

(1) For the supply-side decomposition:

$$GEG = \sum_{i=1}^{N} TI \times \frac{TI_i}{TI} \times \frac{GEG_i}{TI_i} = \sum_{i=1}^{N} TI \times S_{TI_i} \times Eff_{TI_i}, \tag{1}$$

$$NEG = \sum_{j=1}^{M} GC \times \frac{GC_j}{GC} \times \frac{GEG_j}{GC_j \times 8760} \times \frac{NEG_j}{GEG_j} = \sum_{j=1}^{M} GC \times S_{GC_J} \times CUF_J \times Eff_{NEG_j}, \tag{2}$$

(2) For the demand-side decomposition:

$$FEC = \sum_{l=1}^{L} EO \times \frac{EO_l}{EO} \times \frac{FEC_l}{EO_l} = \sum_{l=1}^{L} EO \times S_{EO_l} \times Int_{EO_l}, \tag{3}$$

(3) And an extra equation for balancing the supply and demand sides:

$$FEC = (NEG + Imp - Exp) \times \frac{FEC}{AVEl} = (NEG + Imp - Exp) \times Eff_D, \tag{4}$$

where:

(1) *GEG* is the gross electricity generation; *TI* is the transformation input for electricity generation; $i \ldots N$ is the detailed classification by type of fuel for electricity generation; $S_{TI_i} = \frac{TI_i}{TI}$ is the transformation input structure of electricity generation; and $Eff_{TI_i} = \frac{GEG_i}{TI_i}$ is the transformation efficiency of electricity generation;

(2) *NEG* is the net electricity generation; *GC* is the generation capacity; $j \ldots M$ is the general classification by type of fuel for electricity generation; $S_{GC_J} = \frac{GC_j}{GC}$ is the generation capacity structure; $CUF_J = \frac{GEG_j}{GC_j \times 8760}$ is the capacity utilization factor; and $Eff_{NEG_j} = \frac{NEG_j}{GEG_j}$ is the generation efficiency;

(3) *FEC* is the final electricity consumption; *EO* is the economic output; $l \ldots L$ is the classification by type of economic activity; $S_{EO_l} = \frac{EO_l}{EO}$ is the structure of economic activity; and $Int_{EO_l} = \frac{FEC_l}{EO_l}$ is the intensity of electricity consumption.

(4) *AVEl* is the available electricity, which can be found as $NEG + Imp - Exp$; *Imp* is the import of electricity; *Exp* is the export of electricity; and $Eff_D = \frac{FEC}{AVEl}$ is the distribution efficiency.

As can be seen, each of the equations, except (4), includes extensive, structural, and intensive impacts. Equation (4) has 3 extensive and 1 intensive impact.

Based on the LMDI-I method proposed by B. W. Ang [36,38,39,46], we applied the following models for decomposition analysis:

$$ImpactE = \frac{EF_{i,t} - EF_{i,t-1}}{(\ln EF_{i,t} - \ln EF_{i,t-1})} \times \ln \frac{IF_{i,t}}{IF_{i,t-1}}, \tag{5}$$

$$ImpactP = \exp\left( \frac{EF_{i,t} - EF_{i,t-1}}{(\ln EF_{i,t} - \ln EF_{i,t-1})} \bigg/ \frac{EF_t - EF_{t-1}}{(\ln EF_t - \ln EF_{t-1})} \times \ln \frac{IF_{i,t}}{IF_{i,t-1}} \right), \tag{6}$$

where *Impact*, *GWh*, and *Impact*, % are the impacts of extensive, structural, and intensive factors in absolute and relative terms; $EF_{i,t}$ and $EF_{i,t-1}$ are the electricity flows from individual sources in the current and previous year (*GEG*, *NEG*, and *FEC*); $EF_t$ and $EF_{t-1}$ are the general electricity flows in the current and previous year; and $IF_{i,t}$ and $IF_{i,t-1}$ are the impact factors in the current and previous year.

Conducting a large-scale decomposition of the EU's electric power sector required applying the software package Microsoft Power BI (developed by IBM Corp; [86]) for big data analysis and visualization. As a result of this analysis, the effect of each of the 14 impact factors on changes in the EU's electric power sector was determined (Table 2).

**Table 2.** Factors effecting the fundamental shifts in the EU's electric power sector.

| Impacted Factor | Impacting Factor | Calculation of Impact in Absolute Terms (GWh) |
|---|---|---|
| Gross electricity generation | Input volume impact | $\frac{GEG_{i,t}-GEG_{i,t-1}}{(\ln GEG_{i,t}-\ln GEG_{i,t-1})}\times\ln\frac{TI_t}{TI_{t-1}}$ |
| | Input structure impact | $\frac{GEG_{i,t}-GEG_{i,t-1}}{(\ln GEG_{i,t}-\ln GEG_{i,t-1})}\times\ln\frac{S_{TI_it}}{S_{TI_it-1}}$ |
| | Transformation efficiency impact | $\frac{GEG_{i,t}-GEG_{i,t-1}}{(\ln GEG_{i,t}-\ln GEG_{i,t-1})}\times\ln\frac{Eff_{TI_it}}{Eff_{TI_it-1}}$ |
| Net electricity generation | Generation capacity impact | $\frac{NEG_{j,t}-NEG_{j,t-1}}{(\ln NEG_{j,t}-\ln NEG_{j,t-1})}\times\ln\frac{GC_t}{GC_{t-1}}$ |
| | Generation capacity structure impact | $\frac{NEG_{j,t}-NEG_{j,t-1}}{(\ln NEG_{j,t}-\ln NEG_{j,t-1})}\times\ln\frac{CUF_{Jt}}{CUF_{Jt-1}}$ |
| | Capacity utilization factor impact | $\frac{NEG_{j,t}-NEG_{j,t-1}}{(\ln NEG_{j,t}-\ln NEG_{j,t-1})}\times\ln\frac{CUF_{Jt}}{CUF_{Jt-1}}$ |
| | Generation efficiency impact | $\frac{NEG_{j,t}-NEG_{j,t-1}}{(\ln NEG_{j,t}-\ln NEG_{j,t-1})}\times\ln\frac{Eff_{NEG_jt}}{Eff_{NEG_jt-1}}$ |
| Final electricity consumption | Net electricity generation impact | $\frac{FEC_t-FEC_{t-1}}{(\ln FEC_t-\ln FEC_{t-1})}\times\ln\frac{NEG_t}{NEG_{t-1}}$ |
| | Import impact | $\frac{FEC_t-FEC_{t-1}}{(\ln FEC_t-\ln FEC_{t-1})}\times\ln\frac{Imp_t}{Imp_{t-1}}$ |
| | Export impact | $\frac{FEC_t-FEC_{t-1}}{(\ln FEC_t-\ln FEC_{t-1})}\times\ln\frac{Exp_t}{Exp_{t-1}}$ |
| | Distribution efficiency impact | $\frac{FEC_t-FEC_{t-1}}{(\ln FEC_t-\ln FEC_{t-1})}\times\ln\frac{Eff_{Dt}}{Eff_{Dt-1}}$ |
| | Output volume impact | $\frac{FEC_{l,t}-FEC_{l,t-1}}{(\ln FEC_{l,t}-\ln FEC_{l,t-1})}\times\ln\frac{EO_t}{EO_{t-1}}$ |
| | Output structure impact | $\frac{FEC_{l,t}-FEC_{l,t-1}}{(\ln FEC_{l,t}-\ln FEC_{l,t-1})}\times\ln\frac{S_{EO_lt}}{S_{EO_lt-1}}$ |
| | Consumption intensity impact | $\frac{FEC_{l,t}-FEC_{l,t-1}}{(\ln FEC_{l,t}-\ln FEC_{l,t-1})}\times\ln\frac{Int_{EO_lt}}{Int_{EO_lt-1}}$ |

These impact factors included: (i) extensive factors, which reflect changes in input or output volumes; structural factors, which aim to determine structural shifts in the input or output; (ii) intensive (efficiency) factors, which reflect the available useful electricity obtained after transformation, generation, and distribution, as well as the electricity consumed. As regards efficiency, it is important to note that the rest of the share up to 100% is a measure of inefficiency, and such values should be considered as transformation, generation, and distribution losses, respectively. The contribution of each country to the fundamental shifts in the EU's electric power sector development was determined as the impact of a specific factor for an individual country on the general impact of this factor for the EU as a whole. The main limitation of this research was the lack of a unified classification of electricity generation types for gross electricity generation, net electricity generation, and generation capacities, which made it impossible to deepen the study. Data analysis [79–81] in line with Sankey diagrams [78] will allow for a more detailed decomposition and new conclusions to be drawn regarding the fundamental shifts in the EU's electric power sector.

## 4. Results

First of all, we constructed Sankey diagrams, which allowed us to visualize the changes in the EU's electric power sector in 1995–2021 (Figure 1) and served as the starting point for a further study to explain their causes.

In order to explain the impact of individual factors on the EU's electric power sector, LMDI decomposition was carried out by the stages of electricity flows, which are sequentially presented below.

The LMDI decomposition of gross electricity generation by transformation inputs is presented in Figure 2; the distribution of these impacts across the EU member states is shown in Appendix C.

In 1996–2021, the EU's gross electricity generation grew by 612 TWh (21.2%). Moreover, in 1995–2008, there was a steep upward trend, which resulted in an increase of 637 TWh (22.1%), whereas in 2008–2021, a fluctuating trend was observed which caused a total decrease in its level by −26 TWh (−0.2%). The largest drops in the EU's gross electricity generation occurred in the crisis years of 2009 and 2020: −154 TWh (−4.5%) and −134 TWh

(−4.0%), respectively. However, in the subsequent recovery years, 2010 and 2021, increases of 184 TWh (5.7%) and 147 TWh (4.6%), respectively, were observed.

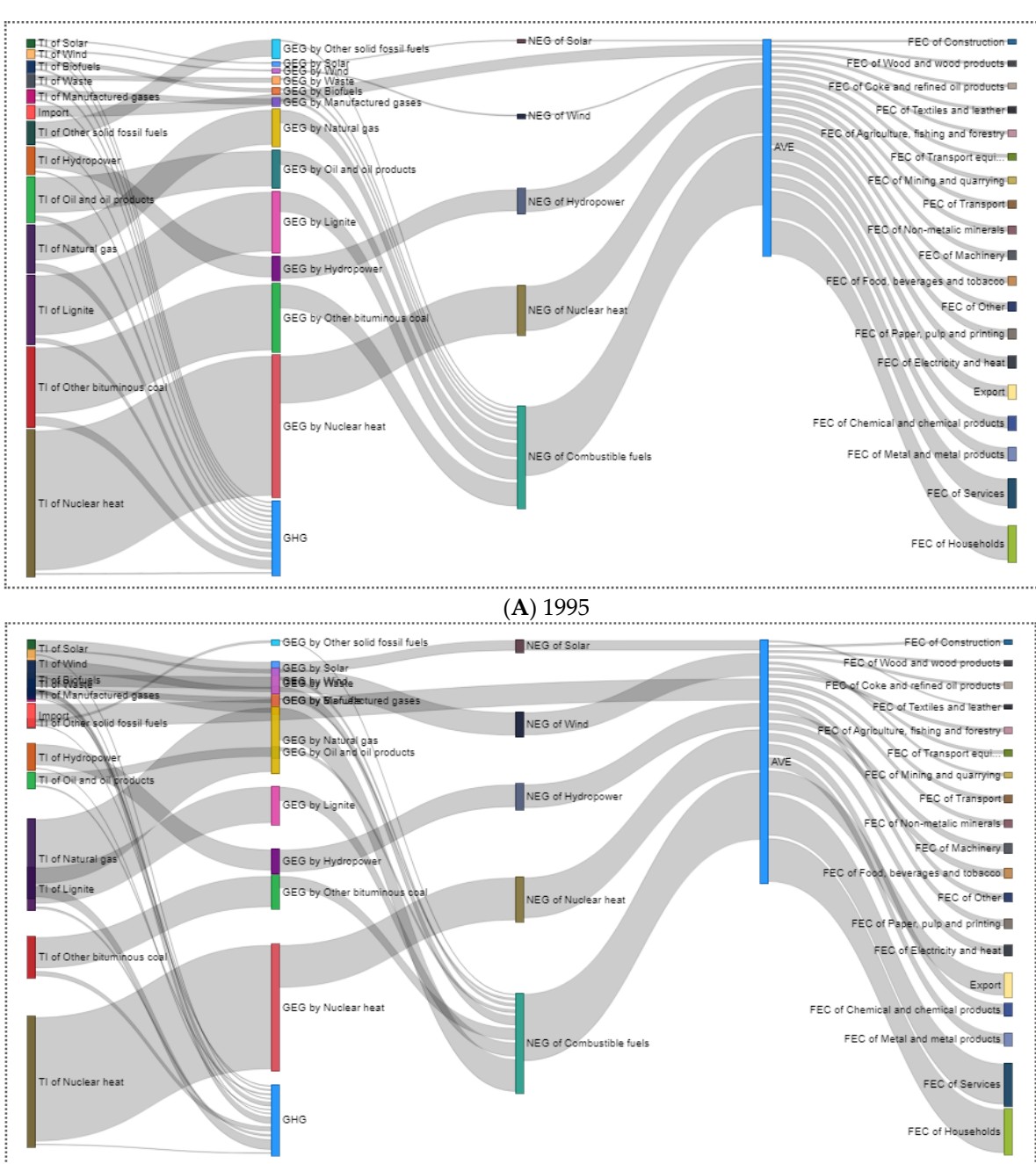

**(A)** 1995

**(B)** 2021

**Figure 1.** Comparison of the Sankey diagrams of the EU's electric power sector for 1995 and 2020. Source: Calculations by the authors based on [78–80].

The pursuit of responsible consumption of energy resources resulted in a reduction in the transformation inputs for the electric power sector (extensive impact factor), which could have led to a decrease in gross electricity generation of −134 TWh (−1.7%). At the same time, we can see that, in 1995–2006, there was an increase in the volume of transformation inputs that caused an increase in gross electricity generation of 479 TWh (15.8%), whereas in 2007–2021, a negative impact of this factor on the trends of gross power generation, which could have led to a decrease of −612 TWh (−17.5%), has already been noted. However, this did not happen due to structural and intensive factors. It was only

in the after-crisis years, 2010 and 2021, that there occurred additional accumulations of transformation inputs to meet the electricity generation demand: +199 TWh (3.6%) and 183 TWh (+5.8%), respectively.

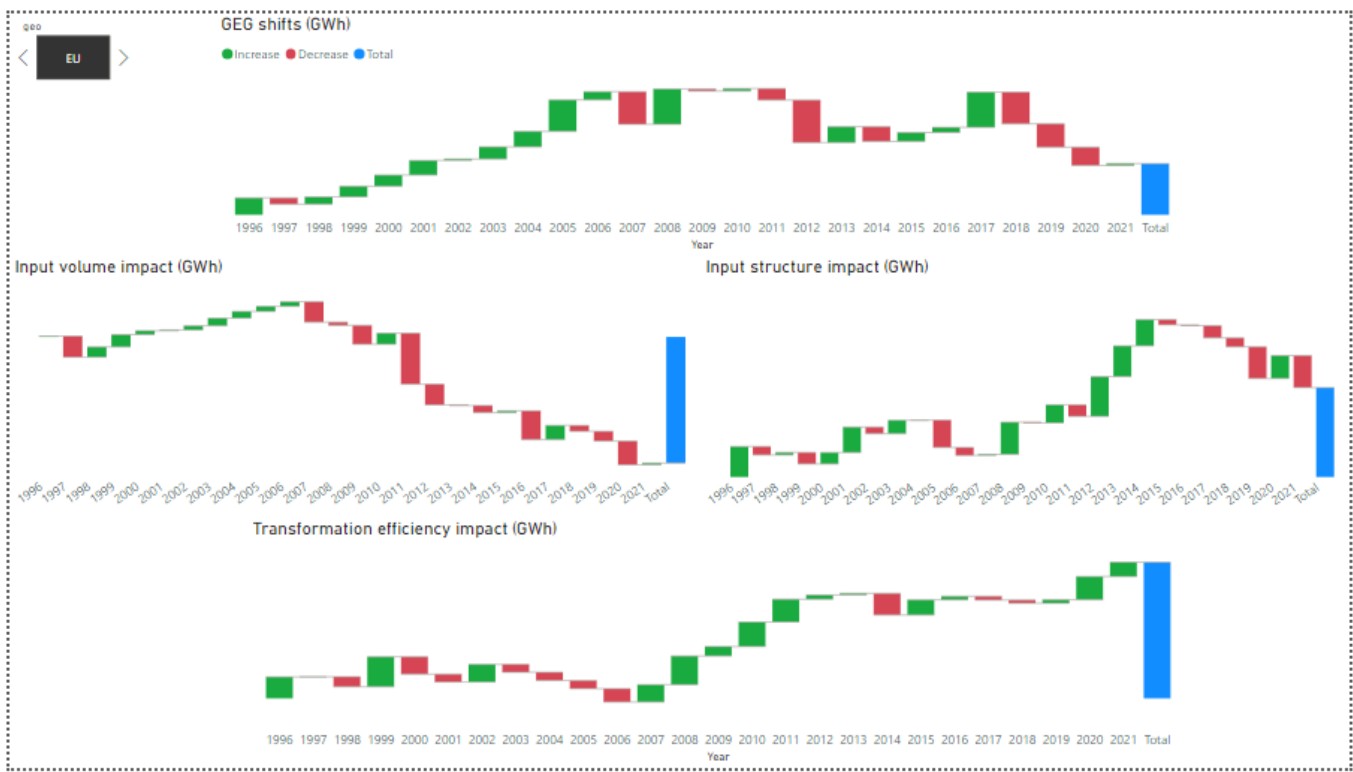

**Figure 2.** Decomposition analysis of the EU's gross electricity generation in 1995–2021. Source: Calculations by the authors based on [78,79].

The increase in the EU's gross electricity generation was mainly due to shifts in the structure of its transformation inputs (structural impact factor), which made it possible to generate additional 600 TWh (+19.2%) in 1995–2021. At the same time, throughout the studied period, these changes caused positive shifts in the structure of gross electricity generation, and only during some years, 2002, 2011, 2014, and 2021, unfavorable market conditions caused structural shifts that negatively affected the EU's gross electricity generation.

The development of electricity generation technologies in the EU also made it possible to achieve positive shifts in transformation efficiency (intensive impact factor). In general, the EU's transformation efficiency grew from 41.8% in 1995 to 53.5% in 2021, which resulted in an increase in its gross electricity generation of +145 TWh (+4.6%). At the same time, although there was an overall positive effect of transformation efficiency on gross electricity generation, the impact of this factor showed a volatile trend.

Thus, it can be stated that the increase in gross electricity generation was mainly caused by changes in the structure of transformation inputs (98%), while transformation efficiency and input volumes exerted a mutually compensating effect, and only a small part of the increase in transformation efficiency (2%) caused a growth in gross electricity generation.

The largest increase in the EU's gross electricity generation was provided by countries such as Spain (+55.5%, or 18.4% of the EU total), Germany (+19.2%, or 18% of the EU total), Italy (+39.1%, or 16.8% of the EU total), France (+17.2%, or 13.9% of the EU total), and Sweden (+26.9%, or 6.8% of the EU total). The only countries that decreased their gross electric electricity generation were Romania (by −47.2%, or −7.9% of the EU total), Lithuania (by −5.4%, or −2.0% of the EU total), Bulgaria (by −3.6%, or −0.8% of the EU

total), Denmark (by −8.6%, or −0.4% of the EU total), and Hungary (by −0.2%, or −0.3% of the EU total). All countries that reduced their demand for transformation inputs also cut the amount of their gross electricity generation, while all countries that developed their electricity generation also required additional transformation inputs, excluding Germany, Greece, Sweden, Slovenia, the Slovak Republic, and Estonia. The greatest positive impacts of the structural shifts on gross electricity generation were achieved by Germany (+34%), Spain (+49%), France (+9%), Italy (+15%), and Sweden (+26%). The transformation input structure had a negative impact on gross electricity generation only in Romania (−3%), the Slovak Republic (−0.6%), the Czech Republic (−0.8%), and Latvia (−2%). The largest increases in gross electricity generation due to growth in transformation efficiency were observed in Italy (+16%), Germany (+7%), the Netherlands (+12%), Austria (+20%), and Belgium (+13%). Transformation efficiency had a negative impact on gross electricity generation in the following countries: Spain (−2%), France (−0.5%), and Estonia (−6%).

The decomposition analysis by type of transformation input allowed us to determine their individual impacts on gross electricity generation (Table 3).

**Table 3.** General impact of transformation inputs on the EU's gross electricity generation in 1995–2021.

| Energy Source | Input Structure Impact (%) | | Transformation Efficiency Impact (%) | |
|---|---|---|---|---|
| | Growth Rate | Share from the EU | Growth Rate | Share from the EU |
| Natural gas | 11.90 | 60.77 | 2.80 | 60.40 |
| Lignite | −4.60 | −24.89 | 0.60 | 11.85 |
| Other bituminous coal | −8.50 | −47.22 | 0.40 | 9.99 |
| Waste | 2.20 | 11.41 | 0.40 | 9.09 |
| Nuclear heat | −1.70 | −9.61 | 0.30 | 6.20 |
| Biofuels | 7.60 | 40.50 | 0.20 | 5.44 |
| Hydro | 1.20 | 5.69 | 0.10 | 2.41 |
| Solar | 5.20 | 28.92 | 0.10 | 1.86 |
| Manufactured gases | 0.00 | 0.80 | 0.10 | 1.79 |
| Other solid fossil fuels | 0.00 | −0.20 | 0.10 | 1.17 |
| Wind | 12.60 | 70.09 | 0.00 | 0.21 |
| Oil and oil products | −6.70 | −36.26 | −0.50 | −10.40 |

Source: Calculations by the authors based on [78,79].

Wind energy had the greatest positive impact on the EU's gross electricity generation. The increase in its share from 0.1% in 1995 to 6.2% in 2021 allowed an increase in gross electricity generation of 420 TWh. At the same time, gas-fired power generation grew from 9.5% in 1995 to 17.9% in 2021, providing an additional 364 TWh of gross electricity generation. The share of biofuels also increased 10-fold, from 0.7% to 7.1%, resulting in an increase in gross electricity generation of 243 TWh. In total, RESs provided an additional 871 TWh (+26.6%) of gross electricity generation, while fossil fuels caused a decrease of 708 TWh (−21.5%). Among the types of transformation inputs, the greatest influence on the growth in gross electricity generation through transformation efficiency was exerted by natural gas (+88 TWh, or +2.8%), the transformation efficiency of oil-based electricity generation had a negative impact, and the individual contributions of the others amounted to less than 1 %. Thus, in 1995–2021, the wind–gas transition of the EU's electric power sector took place. The transition was marked by abandoning fossil fuel sources in favor of wind generation along with the development and improvement of the efficiency of gas-fired generation.

Figure 3 presents the decomposition of the EU's net electricity generation by type of generation capacity; the distribution of the impacts across EU member states is shown in Appendix C.

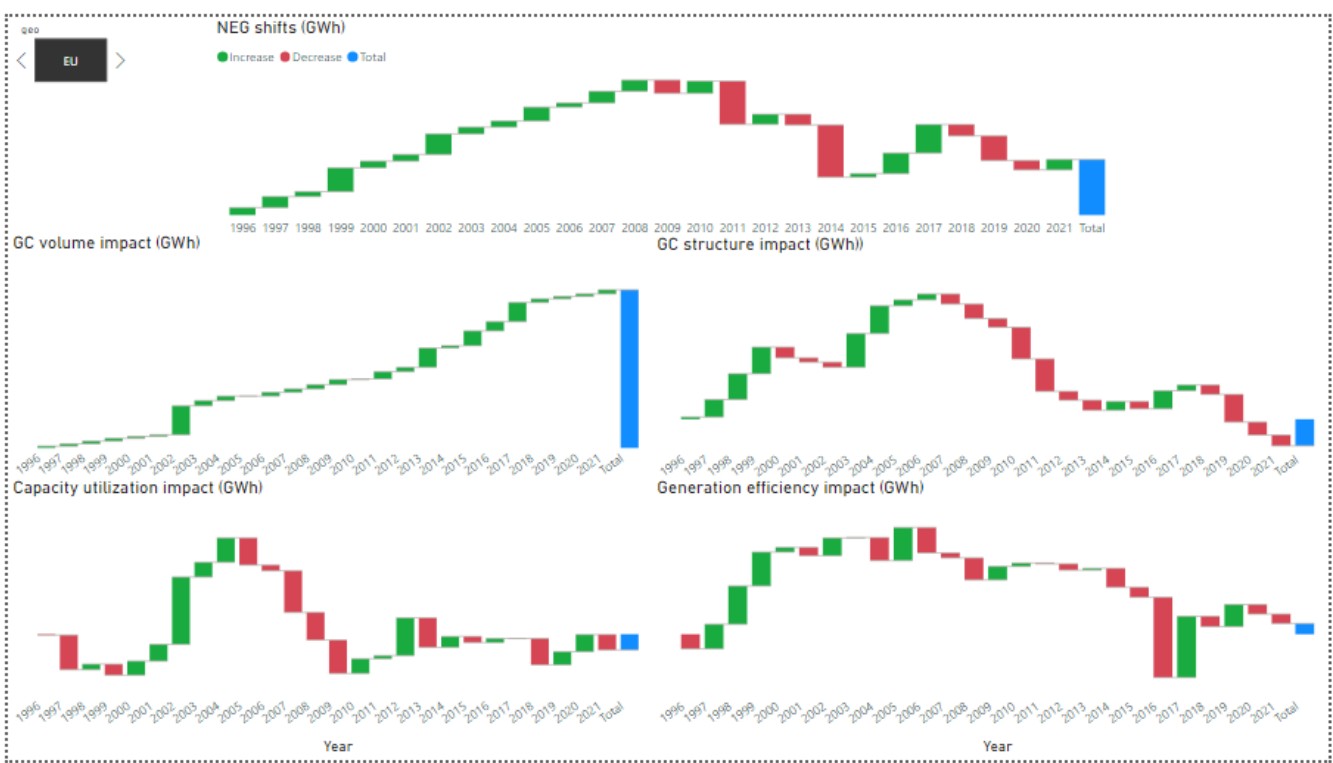

**Figure 3.** Decomposition analysis of the EU's net electricity generation in 1995–2021. Source: Calculations by the authors based on [80,81].

In 1995–2021, the EU's net electricity generation grew by 515 TWh (21.1%), with an increase of 577 TWh (22.8%) in 1995–2008 and a decrease of −63 TWh (−1.7%) in 2009–2021. The reason for this growth was an increase in generation capacities of 431 GW (+57.9%), which could have provided an increase in the net electricity generation of 1556 TWh. However, structural shifts in generation capacities had a negative impact, leading to a decrease in the net electricity generation of −766 TWh (−27%) Similarly, the CUF exerted a negative impact on net electricity generation, causing it to drop by −199 TWh (−5.7%). However, before 2005, the CUF contributed to an increase in the net electricity generation of 121 TWh (+4.9%), while after 2006 a variable negative impact was observed which caused a decline of −319 TWh (−12%). Generation efficiency also exerted a negative impact, causing a decrease of −75 TWh (−2.5%). Before 2007, the increase in generation efficiency resulted in an increase in net electricity generation of 53 TWh (2%), whereas after 2008 there was a drop in net electricity generation by −128 TWh (−4.6%) due to a decline in generation efficiency. Thus, the wind–gas transition of the EU's electric power sector was accompanied by negative shifts in the structure and utilization of its generation capacities and generation efficiency.

Among EU countries, Spain, Germany, and France made the greatest contributions to the growth of the EU's net electricity generation, while only Lithuania showed a reduction in it. The greatest impacts of the increase in the EU's generation capacities on net electricity generation were observed in Germany, Spain, and Italy, while the highest growth rates in net electricity generation for this factor were achieved in Bulgaria, Estonia, and Croatia, with only Lithuania showing negative growth. In all the studied countries, the structural shifts in generation capacities had a negative impact on their net electricity generation, the most powerful occurring in Germany, Bulgaria, and Belgium. The CUF demonstrated different impacts on net electricity generation: the largest decreases were recorded in Spain, the Netherlands, and Poland, while the largest increases were recorded in Bulgaria, Sweden, and Spain. Due to reducing the efficiency of net electricity generation, negative growth was observed in Italy and Lithuania, while the largest drops were recorded in

Estonia and Denmark. The increase in net electricity generation caused positive shifts in Romania, Poland, and the Netherlands. Thus, although the influence of countries with large economies on the increase in net electricity generation was the most tangible, it was countries with smaller economies that were making the greatest efforts.

The decomposition analysis by generation capacities allowed us to determine their individual impacts on net electricity generation (Table 4).

**Table 4.** Impacts of different types of generation capacities on the net electricity generation in the EU in 1995–2021.

| Type of Capacities | GC Structure Impact (%) | | CUF Impact (%) | | Generation Efficiency Impact (%) | |
|---|---|---|---|---|---|---|
| | Growth Rate | Share from the EU | Growth Rate | Share from the EU | Growth Rate | Share from the EU |
| Combustible | −17.2 | −62.8 | −7.1 | −110.9 | −2.8 | −107.7 |
| Hydropower | −5.8 | −20.5 | −0.9 | −16.1 | 0.2 | 4.1 |
| Nuclear | −19.1 | −69.2 | 0.9 | 8.7 | 0.2 | 5.1 |
| Solar | 4.8 | 17.4 | 0.3 | 3.4 | −0.0 | −0.9 |
| Wind | 9.8 | 35.1 | 1.1 | 14.9 | −0.0 | −0.7 |

Source: Calculations by the authors based on [80,81].

As can be seen from Table 3, the greatest negative impact was exerted by combustible capacities, which caused a total drop in the EU's net electricity generation of −782 TWh. Due to the decommissioning of nuclear capacities, the EU's net electricity generation decreased by −530 TWh. The reduction in the volume and utilization of hydropower capacities resulted in a decline in its net electricity generation by −189 TWh, while the improvement in its efficiency provided an additional 3.5 TWh of electricity. Due to the development of intermittent RES capacities, it became possible to increase the net electricity generation by +439 TWh. However, there was a slight drop in their efficiency, which led to a decrease in net electricity generation by −1 TWh. Thus, the rate and volume of the abandonment of traditional generation capacities outpaced the development of renewable ones.

Figure 4 shows the LMDI decomposition of the EU's electricity balance; Appendix D presents the distribution of these impacts across EU countries.

In 1995–2021, the EU's final electricity consumption increased by 495 TWh (21.5%), including an upward trend in 1995–2008, which provided an increase of 568 TWh (23.7%), whereas in 2009–2021 there was a variable downward trend, which resulted in negative growth of −73 TWh (−2.2%). Therefore, after the 2009 crisis, the EU's final electricity consumption decreased at a faster rate compared to the growth in its electricity generation. In 1995–2021, the EU's net electricity generation provided an increase in its final electricity consumption of 410 TWh (82%). In 1995–2008, its growth resulted in covering 470 TWh, whereas in 2009–2021, by contrast, its reduction caused a decrease in the EU's final electricity consumption of −60 TWh. The import growth caused a volatile upward trend, providing an additional 167 TWh (34%) in the EU's final electricity consumption, while the export impact, by contrast, had a volatile downward trend, reducing the available electricity for final electricity consumption by −156 TWh (−31%). The reduction in distribution losses covered an additional 74 TWh, or 14%, of the EU's total final electricity consumption. However, if before 2011 the increase in distribution efficiency resulted in saving 104 TWh of electricity, after 2012, there was a drop in this factor, which resulted in an additional withdrawal of 30 TWh from the amount available for final electricity consumption.

All EU countries increased their final electricity consumption during the period under study. The largest increases in electricity consumption were observed in Ireland, Croatia, and Spain, ranging from 48% to 66.8%. However, Spain, France, Italy, and Poland covered 57.2% of the total European growth in final electricity consumption. The largest contributors to the EU's additional final electricity demand from their own net electricity generation were Spain, Germany, Italy, France, and the Netherlands, which together covered 60.1% of the EU's total additional demand. The countries where net electricity generation had a

negative impact on the supply of the EU's final electricity consumption included Denmark, Estonia, and Latvia. The increase in electricity imports of 49.1% was provided by France, Finland, Denmark, and Hungary, while the increase in exports of 48.8% was covered by the Netherlands, the Czech Republic, and Belgium. Countries such as Germany and Sweden reduced their electricity import demands, while France, Denmark, and also Germany cut their needs for electricity export. The greatest savings of electricity due to an increase in distribution efficiency were recorded in Finland, France, and the Czech Republic, while the greatest excess consumption of electricity due to a drop in distribution efficiency was observed in Austria, Germany, and Denmark.

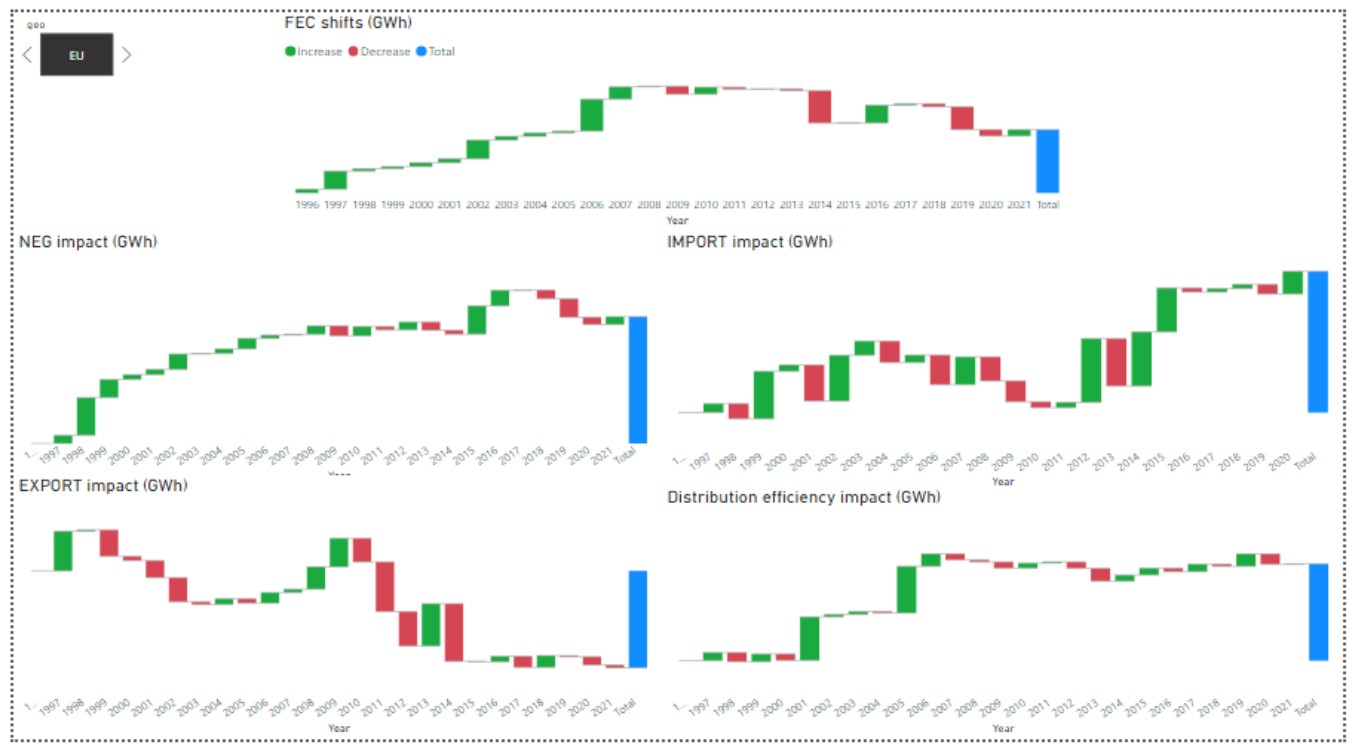

**Figure 4.** Decomposition analysis of the EU's electricity balance in 1995–2021. Source: Calculations by the authors based on [79].

The LMDI decomposition of final electricity consumption by type of economic activity is shown in Figure 5; the distribution of the impacts across EU member states is shown in Appendix F.

The shifts in final electricity consumption shown in Figure 5 caused a 67.1% increase in the EU economy, resulting in an additional 1620 TWh of final electricity consumption, including 1523 TWh (62.8%) in 1995–2008 and 97 TWh (4.3%) in 2009–2021. However, structural shifts in the EU economy resulted in the reduction in this factor by −165 TWh (−13%), and only in certain periods of economic recovery did these structural shifts cause its growth. The decrease in consumption intensity resulted in its reduction by −883 TWh (−34.5%). Thus, one of the driving forces of the EU's energy transition has been responsible final electricity consumption.

Among the EU countries, the largest impact on the growth in final electricity consumption was due to the economic growth of countries with smaller economies, namely, Estonia, Lithuania, Latvia, Romania, and the Slovak Republic, by more than 200%, but on a pan-European scale, the contributions of these countries represented only 12.2%. On the contrary, countries with large economies, such as Germany, France, Spain, and Italy, showed moderate growth in final energy consumption due to their economic development—from 42.1% to 79.8%—but covered 55.4% of the total European growth. The structural shifts in the economy had a mostly negative impact on the decrease in final electricity consumption, the

strongest impacts being observed in Finland and Romania—reaching about −20%—while only in some countries did the structural economic shifts lead to an increase in final electricity consumption, most notably in Croatia and Sweden (about 60%). The decrease in consumption intensity also contributed to a decrease in final electricity consumption in all EU countries. This impact was particularly noticeable in Latvia, Estonia, and Lithuania. However, the Baltic countries' contribution to the EU-wide reduction in final electricity consumption was only −3.5%, while the decreases in consumption intensity in Germany, France, Spain, and Italy ranged from 26.2% to 37.2%, amounting to −61.3% of the total final electricity consumption savings for this factor.

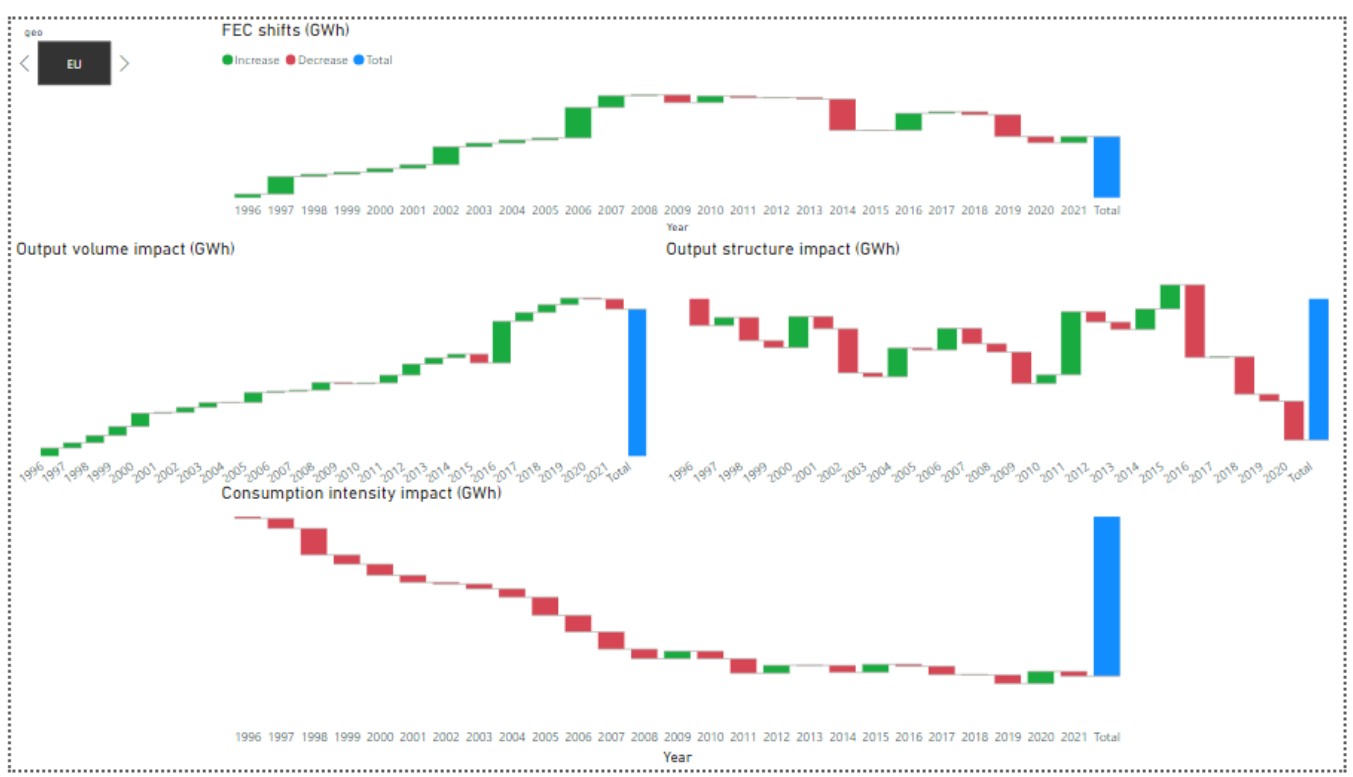

**Figure 5.** Decomposition analysis of the EU's final electricity consumption in 1995–2021. Source: Calculations by the authors based on [79,82,83].

The decomposition analysis conducted by type of economic activity allowed the determination of individual structural and efficiency impacts on the final electricity consumption (Table 5).

As can be seen from Table 5, almost all types of economic activities demonstrated a negative impact on final electricity consumption, most significantly in the chemical and chemical product, the paper, pulp, and printing, and also other sectors, which resulted in electricity savings of −35.3%, −22.9%, and −19.5% of the total EU savings for this factor. The increase in final electricity consumption due to positive structural shifts occurred only in the services, coke and refined oil product, and transport and transport equipment sectors, which together accounted for 36.0% of the growth in the EU's final electricity consumption. As a result of the reduction in consumption intensity, all types of activities showed a decrease in final energy consumption, most significantly households, services, and mining and quarrying, providing, in total, −69.7% of the EU's electricity savings for this factor.

**Table 5.** Impacts of different types of economic activity on the final electricity consumption in the EU in 1995–2021.

| Economic Activity | Impact of Output Structure (%) | | Impact of Output Structure (%) | |
|---|---|---|---|---|
| | Growth Rate | Share from the EU | Growth Rate | Share from the EU |
| Agriculture, fishing, and forestry | −6.0 | −0.5 | −0.3 | −0.1 |
| Chemical and chemical products | −35.3 | −2.7 | −3.1 | −0.7 |
| Coke and refined oil products | 6.2 | 0.5 | −3.8 | −0.8 |
| Construction | −0.2 | 0.0 | −0.5 | −0.1 |
| Food, beverages, and tobacco | −9.5 | −0.7 | −1.1 | −0.2 |
| Households | −8.3 | −0.6 | −33.8 | −7.2 |
| Machinery | −15.5 | −1.2 | 1.8 | 0.4 |
| Metal and metal products | −8.5 | −0.7 | −12.2 | −2.6 |
| Mining and quarrying | −6.8 | −0.5 | −4.2 | −0.9 |
| Non-metallic minerals | −6.5 | −0.5 | −3.4 | −0.7 |
| Other | −19.5 | −1.5 | −1.6 | −0.4 |
| Paper, pulp, and printing | −22.9 | −1.7 | −0.7 | −0.2 |
| Services | 34.8 | 2.7 | −23.7 | −5.1 |
| Textiles and leather | −8.4 | −0.6 | −1.4 | −0.3 |
| Transport | 5.2 | 0.4 | −7.3 | −1.6 |
| Transport equipment | 4.1 | 0.3 | −4.3 | −1.0 |
| Wood and wood products | −3.1 | −0.2 | −0.4 | −0.1 |

Source: Calculations by the authors based on [79,82,83].

## 5. Discussion

The results of the research revealed the following root shifts in the EU's electric power sector development which must be discussed here:

- The EU is undergoing a wind–gas transition in the electric power sector. This has enabled 131% of additional electricity demand to be covered, 70% by wind generation and the rest by gas generation. Overall, the development of renewable generation has provided +145% of additional electricity. At the same time, electricity generation from fossil fuels decreased by −105%, −95% from combustible fossil fuels and −10% from nuclear heat.
- This transition was accompanied by an increase in the energy efficiency of electricity flows, resulting in total electricity savings of 1028 TWh, 85% due to a reduction in the intensity of electricity consumption, while the remaining 15% of the savings came from the increase in the transformation efficiency of inputs in electricity generation. However, this transition led to a decrease in the efficiency of electricity generation and distribution and consequently to overconsumption of electricity by 145 TWh.
- However, this transition was also accompanied by negative structural shifts in the generation capacities and their efficiency, which ultimately led to a lack of 966 TWh in net electricity generation, −50% due to combustible fossil fuel capacities, −55% due to nuclear heat, and −16% due to hydropower capacities, while intermittent RESs (solar and wind) managed to provide only +42% of additional electricity generation.
- The expansion of the EU's economic activity required an increase in its final energy consumption, but the structural shifts in its economy reoriented these volumes from industry (−12%) to households and services and transport (+10%).
- Such changes required an increase in the EU's external electricity flows, which entailed additional electricity exports from net electricity generation (−31%) together with an increase in its electricity imports (+34%) to cover the deficit in final electricity consumption.

Furthermore, the decomposition analysis allowed us to identify critical trends in the development of the EU's electricity sector, among which there are changes in trends in gross and net electricity generation and final electricity consumption (after 2009) and in the impacts of input transformation (from 2007), capacity utilization (from 2006), generation efficiency (from 2008), and distribution efficiency (from 2012).

The current challenges to the EU's power sector development caused by the Russian aggression in Ukraine prove the need to revise the underlying trends of its development. The RePowerEU Plan declares the need for a rapid reduction in the dependence on Russian fossil fuels and acceleration of the energy transition [7]. Consequently, further support for the green energy transition by gas-fired generation is limited. At the same time, the rapid advancement in wind generation requires further enhancement in balancing capacities. The future development of the EU's electric power sector should be based on solar and offshore wind, the deployment of which is ensured by the introduction of electricity storage capacities as well as further electrification of industry and transportation as demand response management systems [7]. However, such future fundamental shifts in the EU's electricity sector require the implementation of important projects of common European interest and their financing through the Innovation Fund [87,88].

## 6. Conclusions

The decomposition of the electric power sector based on the LMDI-I approach can be seen as a powerful methodological tool for studying fundamental shifts in its development, but it requires a lot of manual calculations. In this paper, four LMDI-I models for the decomposition of the fundamental shifts in the EU's electric power sector development have been presented: for gross and net electricity generation, for final electricity consumption, and for balancing supply and demand. In total, the proposed models allowed us to track the electric power sector's development by assessing 14 impact factors (extensive, structural, and intensive) in absolute and relative terms.

In this study, we decomposed the EU's electric power sector by stages of electricity flows for the period 1995–2021 on a large-scale basis (both for the entire EU and for its 25 member states individually) and determined the contribution of each impact factor and each member state to the development of the entire EU's electric power sector. This was achieved by means of big data analysis using the MS Power BI software.

Thus, the designed methodological support for LMDI decomposition allows future fundamental shifts in the development of the electric power sector to be thoroughly and quickly identified and for comparisons with general trends in the EU as a whole and between its member states. Moreover, it can also serve as an analytical tool for regional and state authorities to identify gaps and search for ways to develop the electric power sector.

**Author Contributions:** Conceptualization. V.K. (Viktor Koval) and V.K. (Viktoriia Khaustova); methodology, V.K. (Viktoriia Khaustova), S.L. and T.S.; validation, V.K. (Viktor Koval), S.L. and O.I.; formal analysis, V.K. (Viktoriia Khaustova) and T.S.; project administration, V.K. (Viktoriia Khaustova), V.K. (Viktor Koval) and O.I.; investigation, V.K. (Viktoriia Khaustova) and T.S.; data curation, O.I. and P.O.; writing—original draft preparation, V.K. (Viktoriia Khaustova) and T.S.; writing—review and editing, S.L., O.I. and P.O.; visualization, T.S. and P.O.; supervision, V.K. (Viktor Koval) and V.K. (Viktoriia Khaustova). All authors have read and agreed to the published version of the manuscript.

**Funding:** This research received no external funding.

**Data Availability Statement:** Publicly available datasets were analyzed in this study. The data can be found here: https://ec.europa.eu/eurostat/data/database (accessed on 10 May 2023).

**Conflicts of Interest:** The authors declare no conflict of interest.

## Appendix A

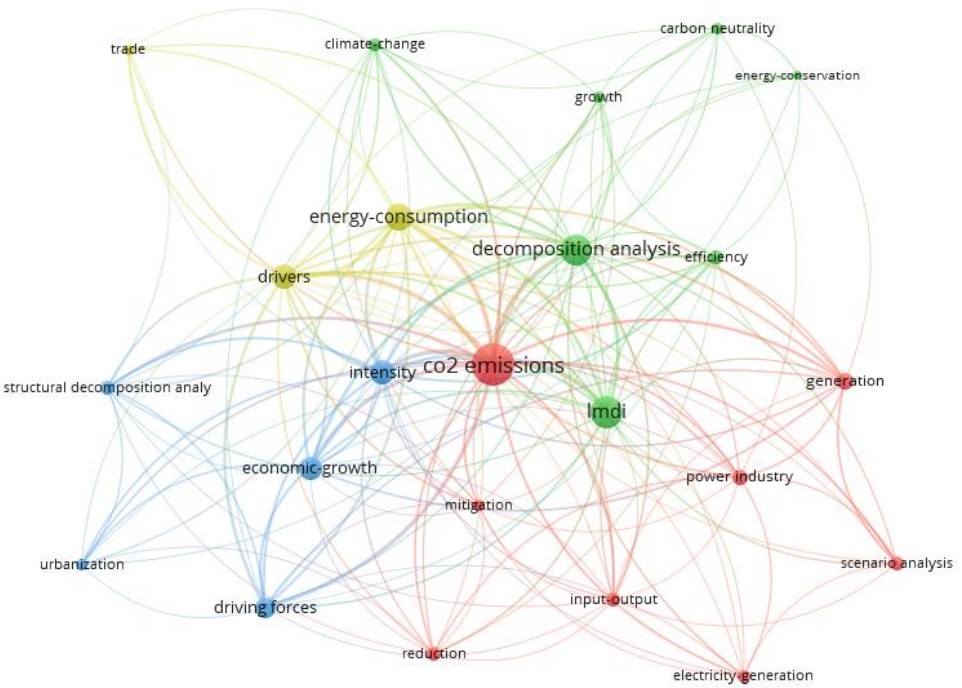

**Figure A1.** Visualization landscape of the scientific problem of "LDMI decomposition of the electric power sector development". Source: Developed by the authors based on [61] with the use of VOSviewer [62].

## Appendix B

**Table A1.** Classification of energy sources for electricity generation decomposition.

| Energy Source (For the Purpose of Gross Electricity Generation) | General Classification of Energy Sources (For the Purpose of Net Electricity Generation) |
|---|---|
| Natural gas | Combustible fuels |
| Lignite | Combustible fuels |
| Other bituminous coal | Combustible fuels |
| Waste | Combustible fuels |
| Nuclear heat | Nuclear heat |
| Biofuels | Nuclear heat |
| Hydro | Hydro |
| Solar | Solar |
| Manufactured gases | Combustible fuels |
| Other solid fossil fuels | Combustible fuels |
| Wind | Wind |
| Oil and oil products | Oil and oil products |

Source: Based on [84].

**Table A2.** Classification of economic activity.

| For Final Electricity Consumption | For Economic Output | Used in Decomposition |
|---|---|---|
| Agriculture and forestry Fishing | Agriculture, fishing, and forestry | Agriculture, fishing, and forestry |
| Chemical and petrochemical | Chemical and chemical products Pharmaceutical products | Chemical and chemical products |
| Coke oven and oil refinery Construction | Coke and refined oil products Construction | Coke and refined oil products Construction |

**Table A2.** *Cont.*

| For Final Electricity Consumption | For Economic Output | Used in Decomposition |
|---|---|---|
| Food, beverages, and tobacco | Food, beverages, and tobacco | Food, beverages, and tobacco |
| Households | Household expenditures | Households |
| Machinery | Machinery | Machinery |
| Iron and steel | Basic metals and fabricated metal products | Metal and metal products |
| Non-ferrous metals | | |
| Mining and quarrying | Mining and quarrying | Mining and quarrying |
| Non-metallic minerals | Non-metallic minerals | Non-metallic minerals |
| Not elsewhere specified | Other | Other |
| Paper, pulp, and printing | Paper and paper products, reproduction | Paper, pulp, and printing |
| Commercial and public services | All services excluding the transport sector | Services |
| Textiles and leather | Textiles and leather | Textiles and leather |
| Transport | Transport sector | Transport |
| Transport equipment | Transport equipment | Transport equipment |
| Wood and wood products | Wood and wood products | Wood and wood products |

Source: Based on [85].

**Appendix C**

**Table A3.** Decomposition analysis of gross electricity generation in the EU member states in 1995–2021.

| Geo [1] | Input Volume Impact (%) | | Input Structure Impact (%) | | Transformation Efficiency Impact (%) | | GEG Shifts (%) | |
|---|---|---|---|---|---|---|---|---|
| | GR [2] | Share [3] | GR | Share | GR | Share | GR | Share |
| AT | 11.4 | −4.5 | 7.5 | 0.5 | 19.5 | 9.9 | 33.4 | 3.7 |
| BE | 21.2 | −5.0 | 19.4 | 1.9 | 12.7 | 8.2 | 40.9 | 4.9 |
| BG | −8.1 | 6.1 | 8.3 | 0.4 | 2.7 | 0.7 | −3.6 | −0.8 |
| CZ | 10.7 | −6.2 | 0.8 | −0.1 | 6.4 | 4.3 | 15.7 | 2.3 |
| DE | −20.3 | 119.9 | 34.1 | 37.3 | 6.9 | 31.4 | 19.2 | 18.0 |
| DK | −16.1 | 18.0 | 30.3 | 2.5 | 11.6 | 4.7 | 8.6 | −0.4 |
| EE | −8.8 | 1.9 | 262.9 | 1.5 | −5.8 | −0.5 | 178.5 | 0.9 |
| EL | −7.1 | 5.7 | 43.9 | 3.4 | 5.0 | 0.8 | 32.6 | 2.3 |
| ES | 20.4 | −17.9 | 49.2 | 17.2 | −1.8 | −7.0 | 55.5 | 18.4 |
| FI | 12.4 | −1.2 | 16.5 | 1.5 | 8.7 | 5.3 | 27.7 | 3.1 |
| FR | 9.1 | −28.3 | 9.3 | 8.1 | 0.5 | −0.7 | 17.2 | 13.9 |
| HR | 43.7 | −3.5 | 25.0 | 0.1 | 9.4 | 1.0 | 57.3 | 1.1 |
| HU | −16.6 | 6.3 | 8.5 | 0.3 | 9.5 | 2.7 | −0.2 | −0.3 |
| IE | 23.2 | −2.6 | 39.2 | 1.4 | 16.0 | 2.6 | 72.4 | 2.6 |
| IT | 10.6 | −6.9 | 15.1 | 7.2 | 16.4 | 34.7 | 39.1 | 16.8 |
| LT | −91.0 | 17.5 | 132.9 | 2.7 | 12.9 | 0.5 | −54.1 | −2.0 |
| LU | 200.1 | −0.8 | 73.2 | 0.1 | 43.9 | 0.9 | 179.2 | 0.4 |
| LV | 23.5 | −1.0 | 21.0 | 0.0 | 5.7 | 0.3 | 30.8 | 0.3 |
| NL | 12.3 | −8.1 | 9.8 | 1.4 | 11.6 | 10.5 | 29.9 | 5.7 |
| PL | 6.1 | −7.3 | 8.5 | 2.8 | 2.9 | 3.9 | 16.4 | 5.3 |
| PT | 16.4 | −0.8 | 58.5 | 2.8 | 10.1 | 2.7 | 56.4 | 3.6 |
| RO | −46.2 | 34.8 | 3.0 | −0.7 | 3.8 | 1.8 | −47.2 | −7.9 |
| SE | 6.7 | 0.2 | 26.1 | 6.5 | 1.3 | 1.8 | 26.9 | 6.8 |
| SK | 2.2 | 0.2 | −0.6 | −0.3 | 10.2 | 2.4 | 9.0 | 0.3 |
| SL | 3.9 | 0.2 | 14.5 | 0.3 | 5.1 | 0.6 | 20.3 | 0.4 |

Source: Calculations by the authors based on [80,81]. Notes: Here and after: [1] Country abbreviations are used according to ISO 3166-1 alpha-2 [87,88]; [2] GR = growth rate; [3] Share = the share of the total EU impact.

## Appendix D

**Table A4.** Decomposition analysis net electricity generation in the EU member states in 1995–2021.

| Geo | GC Impact (%) | | GC Structure Impact (%) | | CUF Impact (%) | | Generation Efficiency Impact (%) | | NEG Shifts (%) | |
|-----|------|-------|------|-------|------|-------|------|-------|------|-------|
| | GR | Share | GR | Share | GR | Share | GR | Share | GR | Share |
| AT | 46.3 | 1.9 | −11.6 | 1.1 | 2.6 | 1.1 | −6.7 | 5.7 | 27.4 | 3.1 |
| BE | 57.9 | 3.1 | −33.9 | 3.8 | 23.5 | −5.4 | −4.3 | 4.9 | 39.5 | 5.0 |
| BG | 415.4 | 4.3 | −43.1 | 2.4 | 53.7 | −9.5 | 16.7 | −7.7 | 20.5 | 1.2 |
| CZ | 43.6 | 1.9 | −5.7 | 0.7 | −9.7 | 4.3 | 9.9 | −7.6 | 35.3 | 4.4 |
| DE | 75.2 | 27.8 | −51.7 | 40.0 | 2.3 | 6.9 | −8.6 | 70.3 | 12.2 | 11.4 |
| DK | 43.2 | 0.9 | −13.2 | 0.7 | 11.4 | 0.4 | −28.9 | 15.0 | 13.8 | −0.5 |
| EE | 286.0 | 2.9 | −15.8 | 0.2 | 198.7 | −7.0 | −101.3 | 18.6 | 20.0 | −0.2 |
| EL | 86.2 | 2.8 | −19.0 | 1.4 | −25.2 | 9.0 | 0.5 | 0.1 | 36.0 | 2.8 |
| ES | 91.9 | 14.7 | −19.0 | 6.7 | −8.9 | 31.9 | −1.6 | 6.5 | 53.7 | 20.9 |
| FI | 22.9 | 0.9 | −5.3 | 0.5 | 18.7 | −4.6 | −14.3 | 14.5 | 19.3 | 1.7 |
| FR | 28.1 | 9.4 | −15.7 | 11.3 | 4.3 | −6.0 | −2.5 | 17.9 | 12.7 | 11.4 |
| HR | 111.0 | 0.7 | −38.5 | 0.8 | 47.1 | −1.2 | 6.9 | −0.7 | 69.7 | 1.1 |
| HU | 46.7 | 0.9 | −20.4 | 0.9 | −17.4 | 3.6 | 6.7 | −2.6 | 12.6 | 0.5 |
| IE | 104.6 | 1.7 | −12.0 | 0.5 | −16.8 | 3.3 | −4.9 | 2.0 | 63.3 | 2.8 |
| IT | 59.3 | 10.4 | −16.7 | 6.4 | 2.2 | 4.2 | −18.9 | 77.3 | 19.6 | 9.0 |
| LT | −37.3 | −0.4 | −0.6 | 0.5 | 63.1 | −4.4 | 18.9 | −2.9 | −18.8 | −1.4 |
| LU | 47.3 | 0.1 | 63.3 | −0.1 | 25.1 | 0.1 | −25.3 | 1.4 | 98.8 | 0.2 |
| LV | 36.9 | 0.1 | 1.6 | 0.0 | 15.1 | 0.9 | 47.8 | −2.7 | 87.5 | 0.4 |
| NL | 94.2 | 6.1 | −37.8 | 5.7 | −14.0 | 10.2 | 8.4 | −11.8 | 44.4 | 7.8 |
| PL | 60.4 | 5.6 | −20.8 | 4.2 | −16.1 | 13.6 | 7.6 | −14.2 | 27.5 | 7.4 |
| PT | 85.1 | 2.5 | −10.1 | 0.8 | 14.5 | 4.4 | −8.4 | 5.1 | 54.2 | 3.8 |
| RO | 14.1 | 0.4 | −5.0 | 0.4 | −23.3 | 7.9 | 27.8 | −18.2 | 7.7 | 0.4 |
| SE | 29.6 | 2.8 | −21.9 | 4.6 | 18.0 | −8.6 | −0.5 | 1.4 | 21.3 | 4.7 |
| SK | 5.7 | 0.0 | −2.1 | 0.2 | 13.5 | −1.6 | 7.5 | −2.2 | 19.1 | 0.8 |
| SL | 49.6 | 0.5 | −14.7 | 0.3 | −8.1 | 1.0 | 6.9 | −1.0 | 30.0 | 0.7 |

Source: Calculations by the authors based on [80,81].

## Appendix E

**Table A5.** Decomposition analysis of the electricity balance in the EU member states in 1995–2021.

| Geo | NEG Impact (%) | | IMPORT Impact (%) | | EXPORT Impact (%) | | Distribution Efficiency Impact, (%) | FEC Shifts (%) | |
|-----|------|-------|------|-------|------|-------|------|------|-------|
| | GR | Share | GR | Share | GR | Share | GR | GR | Share |
| AT | 30.7 | 3.90 | 9.8 | 9.20 | −7.8 | 6.10 | −51.80 | 36.0 | 4.2 |
| BE | 27.2 | 5.60 | 4.9 | 3.20 | −2.9 | 11.00 | 8.70 | 17.4 | 2.7 |
| BG | 27.9 | 1.20 | 1.5 | 1.00 | −5.9 | 6.30 | 9.20 | 7.4 | 0.4 |
| CZ | 32.9 | 4.70 | 4.5 | 3.70 | −9.8 | 11.00 | 33.50 | 20.1 | 2.4 |
| DE | 28.5 | 10.50 | 1.9 | −4.30 | −2.6 | −0.30 | −45.50 | 5.4 | 4.9 |
| DK | 25.9 | −4.60 | 8.4 | 9.40 | −8.4 | −2.70 | −50.00 | 1.5 | 0.1 |
| EE | 32.1 | −0.40 | 7.4 | 4.20 | −10.9 | 2.40 | 15.40 | 41.6 | 0.6 |
| EL | 32.6 | 3.10 | 4.2 | 2.80 | −1.3 | 1.60 | −7.50 | 31.9 | 2.7 |
| ES | 39.6 | 22.40 | 1.8 | 5.20 | −1.4 | 6.00 | −17.00 | 48.0 | 17.9 |
| FI | 27.4 | 0.70 | 3.5 | 9.60 | −1.2 | 3.40 | 53.80 | 23.8 | 3.4 |
| FR | 33.2 | 8.80 | 0.7 | 11.50 | −4.1 | −1.90 | 36.50 | 22.6 | 17.8 |
| HR | 30.0 | 1.00 | 19.0 | 4.60 | −10.2 | 3.50 | 7.30 | 50.2 | 1.3 |
| HU | 24.1 | 0.40 | 10.4 | 9.30 | −4.8 | 3.50 | 26.00 | 36.3 | 2.8 |
| IE | 38.8 | 3.00 | 1.8 | 1.40 | −0.8 | 0.40 | −7.10 | 66.8 | 2.9 |
| IT | 25.9 | 9.70 | 4.3 | 4.80 | −0.2 | 1.90 | 11.40 | 19.5 | 10.5 |
| LT | 18.2 | −2.20 | 6.9 | 3.10 | −4.0 | 0.50 | 18.50 | 40.3 | 0.8 |
| LU | 13.7 | 0.20 | 24.4 | 0.40 | −7.1 | 0.20 | 2.20 | 26.2 | 0.3 |
| LV | 25.1 | 0.80 | 15.0 | 1.20 | −9.9 | 1.80 | −7.30 | 37.0 | 0.4 |

**Table A5.** *Cont.*

| Geo | NEG Impact (%) | | IMPORT Impact (%) | | EXPORT Impact (%) | | Distribution Efficiency Impact, (%) | FEC Shifts (%) | |
|---|---|---|---|---|---|---|---|---|---|
| | GR | Share | GR | Share | GR | Share | GR | GR | Share |
| NL | 28.4 | 8.70 | 6.4 | 5.60 | −3.1 | 12.50 | −13.70 | 29.0 | 5.8 |
| PL | 28.6 | 8.10 | 1.7 | 6.20 | −2.2 | 3.70 | 24.40 | 40.6 | 10.9 |
| PT | 36.9 | 4.20 | 5.4 | 3.00 | −3.3 | 1.00 | −9.40 | 47.5 | 3.6 |
| RO | 26.0 | 0.50 | 1.3 | 4.90 | −2.2 | 3.80 | −11.20 | 13.7 | 1.1 |
| SE | 26.4 | 7.30 | 2.4 | −4.20 | −3.7 | 14.30 | 20.00 | 2.7 | 0.6 |
| SK | 35.8 | 1.10 | 15.9 | 4.90 | −18.4 | 7.00 | 3.70 | 14.3 | 0.7 |
| SL | 26.5 | 0.80 | 10.1 | 4.40 | −9.8 | 3.90 | −20.40 | 32.3 | 0.8 |

Source: Calculations by the authors based on [79].

**Appendix F**

**Table A6.** Decomposition analysis of the final electricity generation in the EU member states in 1995–2021.

| Geo | Output Volume Impact (%) | | Output Structure Impact (%) | | Consumption Intensity Impact (%) | | FEC Shifts (%) | |
|---|---|---|---|---|---|---|---|---|
| | GR | Share | GR | Share | GR | Share | GR | Share |
| AT | 88.2 | 3.3 | −4.3 | −6.4 | −41.9 | −3.1 | 36.0 | 4.2 |
| BE | 86.7 | 4.4 | −2.6 | −6.8 | −46.9 | −4.8 | 17.4 | 2.7 |
| BG | 181.5 | 3.2 | 4.1 | 3.1 | −113.8 | −4.1 | 7.4 | 0.4 |
| CZ | 178.0 | 6.3 | −6.2 | −10.1 | −108.1 | −7.6 | 20.1 | 2.4 |
| DE | 42.1 | 14.0 | 8.1 | 84.1 | −37.2 | −25.1 | 5.4 | 4.9 |
| DK | 95.4 | 2.0 | −10.4 | −9.8 | −67.8 | −2.6 | 1.5 | 0.1 |
| EE | 247.1 | 1.0 | −8.6 | −1.5 | −132.7 | −1.1 | 41.6 | 0.6 |
| EL | 56.7 | 1.5 | −6.7 | −7.5 | −0.3 | −0.1 | 31.9 | 2.7 |
| ES | 79.8 | 9.6 | 10.0 | 65.6 | −26.2 | −7.5 | 48.0 | 17.9 |
| FI | 96.6 | 4.7 | −22.2 | −50.8 | −38.6 | −3.9 | 23.8 | 3.4 |
| FR | 74.8 | 19.5 | −5.9 | −71.2 | −34.2 | −17.9 | 22.6 | 17.8 |
| HR | 30.7 | −0.4 | 63.9 | 22.3 | −44.2 | −0.7 | 50.2 | 1.3 |
| HU | 158.4 | 3.3 | −7.5 | −6.7 | −79.0 | −3.3 | 36.3 | 2.8 |
| IE | 175.0 | 2.3 | −2.6 | −2.1 | −74.3 | −2.0 | 66.8 | 2.9 |
| IT | 72.3 | 12.3 | −2.1 | −17.5 | −32.7 | −10.8 | 19.5 | 10.5 |
| LT | 236.8 | 1.3 | 9.5 | 3.0 | −131.1 | −1.4 | 40.3 | 0.8 |
| LU | 229.5 | 0.8 | −15.4 | −2.7 | −62.4 | −0.5 | 26.2 | 0.3 |
| LV | 210.5 | 0.7 | 11.5 | 2.2 | −135.2 | −0.9 | 37.0 | 0.4 |
| NL | 95.8 | 6.2 | −9.5 | −27.2 | −40.3 | −5.1 | 29.0 | 5.8 |
| PL | 159.4 | 11.2 | 4.8 | 26.6 | −80.9 | −12.1 | 40.6 | 10.9 |
| PT | 62.6 | 1.5 | 8.8 | 12.2 | −17.0 | −1.1 | 47.5 | 3.6 |
| RO | 224.8 | 6.1 | −20.0 | −25.4 | −103.8 | −5.7 | 13.7 | 1.1 |
| SE | 12.9 | 4.9 | 58.9 | 139.8 | −54.1 | −8.8 | 2.7 | 0.6 |
| SK | 204.2 | 3.2 | −11.4 | −9.1 | −108.9 | −3.4 | 14.3 | 0.7 |
| SL | 115.7 | 0.9 | −4.9 | −1.7 | −40.2 | −0.6 | 32.3 | 0.8 |

Source: Calculations by the authors based on [79,82,83].

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
