# Peer review of "Fundamental Shifts in the EU’s Electric Power Sector Development: LMDI Decomposition Analysis"

_energies, doi:10.3390/en16145478_

Round 1

Reviewer 1 Report

The authors of this paper state that the aim of the paper is to study the shifts of the EU electric power sector development based on the LMDI-decomposition by the stages of electricity flows. The paper is interesting, but requires major improvements:

1.       Please, use a detailed description of the bibliographic sources. The source [8], which is heavily cited, requires specific details. Source [8] is a database, namely Eurostat, and an open data base must be cited for each series employed. Please, modify for all inputs from Eurostat, namely source [8].

2.       Please, improve English language. For instance, within the abstract modify “It was proposed 4 models...” with “ Four models have been proposed….”. Some sentences make no sense, while some words are being repeated or used without any meaning. Major grammar errors make the text difficult to follow; please, make sure you formulate clearly and correctly each statement, not like the following “So, there are lack of deep investigation electric power sector by stages of electricity flows.”

3.       The authors have selected only 25 member states of the EU. Why have you removed Luxembourg, Malta and Cyprus? If there is a lack of data, please make sure you add this limitation.

4.       Please, add more description to the methodology section, describing all the analysed factors.

5.       Please, clearly describe the original contributions to this paper.

6.       Please, better describe the 14 factors which you claim to impact the fundamental shifts in EU electric power sector.

7.       Which are the limitations of the present research?

Dear Editor,

Thnak you for inviting me to review this paper. The paper is interesting, but requires major improvments and extensive editing of English language.

Author Response

Dear Reviewer

Please find attached file

Reviewer 2 Report

I have the following recommendations:

1. The title and the abstract should not have abbreviations, such as LMDI. The abbreviations should be introduced, explained the first time they appear in the text.

2. The introduction and literature review are well structured with sufficient references which are relevant to the issue analysed. The results and the method used are clearly explained. 

3. Discussion and conclusions should be separate. Discussions should include a parallel/comparison between your results and other results found by other researchers. 

4. Conclusions should have theoretical and practical implications for your results (at the national level but also for individuals, and companies), limitations of your research and future research directions. 

5. No need to have a list of abbreviations at the end, you should introduce them the first time appear in the text. 

6. The list of references needs some newer papers (2021-2023). Please add a few in the Discussion or where you consider. But more new references are needed especially considering your topic. 

The text should be checked by a professional, there are phrases that are difficult to read. An example would be lines 70-74. Lines 201-202, after including no need to put the word in and other examples. 

Author Response

Dear Reviewer

Please find attached file

Reviewer 3 Report

The application of LMDI-decomposition is an interesting approach to understand the behaviour of EU electric power sector.

Two comments:

1. On the electric power sector supply chain, the energy losses (in some cases) are very important factor to consider. In this sense, the model described in the section 2, why the energy losses are not considered? 

In the line 350, page 11/23, is mentioned:

"The reduction of distribution losses made it possible to 350 cover an additional 74 TWh (14%) of the total EU final electricity consumption"

2. Please check the Legen (x-axis) of Figure 2, "Input structure impact (GWh)

Author Response

Dear Reviewer

Please find attached file

Round 2

Reviewer 1 Report

The authors addressed the reviewers recommendations.

The authors addressed the reviewers recommendations.